# Predictors of Mental Health Service Utilization among Frontline Healthcare Workers during the COVID-19 Pandemic

**DOI:** 10.3390/ijerph20075326

**Published:** 2023-03-30

**Authors:** Sydney Starkweather, Jonathan M. DePierro, Saadia Akhtar, Eleanore de Guillebon, Carly Kaplan, Sabrina Kaplan, Jonathan Ripp, Lauren Peccoralo, Jordyn Feingold, Adriana Feder, James W. Murrough, Robert H. Pietrzak

**Affiliations:** 1Department of Psychiatry, Icahn School of Medicine at Mount Sinai, One Gustave L. Levy Place, New York, NY 10029, USA; 2Departments of Emergency Medicine and Graduate Medical Education, Icahn School of Medicine at Mount Sinai, One Gustave L. Levy Place, New York, NY 10029, USA; 3Office of Well-Being and Resilience, Icahn School of Medicine at Mount Sinai, One Gustave L. Levy Place, New York, NY 10029, USA; 4Department of Medicine, Icahn School of Medicine at Mount Sinai, One Gustave L. Levy Place, New York, NY 10029, USA; 5U.S. Department of Veterans Affairs National Center for PTSD, West Haven, CT 06516, USA; 6Department of Psychiatry, Yale School of Medicine, 300 George Street, New Haven, CT 06511, USA

**Keywords:** health care workers, mental health, treatment utilization, psychological symptoms, barriers to care, COVID-19, systemic racism, health services

## Abstract

(1) Background: This study examined the prevalence and correlates of factors associated with self-reported mental health service use in a longitudinal cohort of frontline health care workers (FHCWs) providing care to patients with COVID-19 throughout 2020. (2) Methods: The study comprised a two-wave survey (*n* = 780) administered in April–May 2020 (T1) and November 2020–January 2021 (T2) to faculty, staff, and trainees in a large urban medical center. Factors associated with initiation, cessation, or continuation of mental health care over time were examined. (3) Results: A total of 19.1% of FHCWs endorsed currently utilizing mental health services, with 11.4% continuing, 4.2% initiating, and 3.5% ceasing services between T1 and T2. Predisposing and need-related factors, most notably a history of a mental health diagnosis and distress related to systemic racism, predicted service initiation and continuation. Among FHCWs with a prior mental health history, those with greater perceived resilience were less likely to initiate treatment at T2. Descriptive data highlighted the importance of services around basic and safety needs (e.g., reliable access to personal protective equipment) relative to mental health support in the acute phase of the pandemic. (4) Conclusions: Results may be helpful in identifying FHCWs who may benefit from mental health services.

## 1. Introduction

The COVID-19 pandemic brought unprecedented amounts of stress to frontline healthcare workers (FHCWs). Our group found that 39% of 2759 FHCWs at an urban tertiary care hospital in New York City (NYC) screened positive for symptoms of major depression (MDD), generalized anxiety (GAD), and/or posttraumatic stress (PTSD) symptoms in April to May 2020 [1,2]; global studies with FHCWs have shown an even higher incidence of positive screens for psychiatric disorders in 2020 [3]. However, few studies have examined actual mental health service use among FHCWs during the pandemic. One large-scale study of physicians in Canada (*n* = 34,055) found that 13.4% of the sample utilized mental health or substance use services at least once over the first twelve months of the pandemic [4]. A better understanding of the rates of service use, and barriers or facilitators to obtaining care, may drive the development of tailored services.

The Andersen Behavioral Model of Health Services Use (ABMHSU) [5] is a framework for organizing individual and community-level factors associated with engaging in health services. The model includes ‘predisposing’ factors (e.g., demographic characteristics such as age, race, and social factors such as education, occupation, family status, and social support); ‘enabling’ factors (e.g., having accessible and affordable treatment options), and ‘need’ factors (e.g., illness severity, environmental or occupational-related injury) [5]. In this study, the ABMHSU serves as a framework for understanding the individual, contextual, and needs-based factors that may contribute to or inhibit mental health service use by FHCWs during the COVID-19 pandemic. Similar studies have used the ABMHSU as a framework for understanding health service use amongst first responders. For example, in World Trade Center rescue, recovery, and cleanup workers, a combination of demographic and need-based factors, including female gender and current psychiatric distress, respectively, predicted willingness to engage with mental health services if needed [6]. Similarly, a recent study found that veterans who were female and younger in age (predisposing factors) and had higher distress and lifetime trauma burden (need factors) were more likely to utilize services, while predisposing factors alone were largely unrelated to this outcome [7]. This same study categorized protective psychosocial characteristics, such as dispositional optimism and “grit”, as needs factor in the ABMHSU framework because of their potential to affect an individual’s perceived need for resources, and thus potentially inhibit or facilitate service use. To our knowledge, this model has not been applied to identifying factors associated with mental health service use in FHCWs.

FHCWs often faced extreme occupational stressors (typically classified as predisposing factors) related to their COVID-19 response work. Additionally, in parallel to the pandemic, there was a rise in racially motivated violence and increased attention to systemic racism toward Black and Asian Americans [8,9,10], and the disproportionate mortality burden among people of color early in the pandemic [11]. These factors may have impacted the overall levels of stress felt by FHCWs during the COVID-19 pandemic. For instance, one survey found that 30% of healthcare workers were at least moderately distressed by systemic racism in August-September 2020 [12]. To date, however, whether racism-related distress may be associated with mental health treatment utilization among FHCWs has not been evaluated.

This study aims to address gaps in the literature by evaluating factors associated with self-reported initiation, continuation, or cessation of mental health services in a cohort of COVID-19 FHCWs at a large urban medical center followed longitudinally over 2020. Aligned with prior work, we hypothesize that demographic factors, such as female gender identity, prior mental health history, and current psychiatric symptom burden, will emerge as the strongest predictors of service use (inclusive of the initiation and continuation of services across the study period). We also hypothesize that distress related to systemic racism, which can be considered a need factor, will be associated with an increased likelihood of service use. As a secondary aim, we sought to describe services that FHCWs perceived as most helpful in reducing stress during the pandemic, spanning from basic and safety needs to mental health services. 

## 2. Materials and Methods

### 2.1. Participants

Data were collected via two anonymous surveys of FHCWs working at Mount Sinai Hospital (MSH), an urban tertiary care hospital in NYC. The first survey was administered during the middle and downward slope of the initial pandemic peak in NYC in April–May 2020 (Time 1 [T1]), and the second survey was administered at 7-months follow-up between November 2020–January 2021 (Time 2 [T2]), corresponding to a subsequent rise and plateau of the second pandemic surge in NYC (as indicated by inpatient census data).

Surveys were created and administered using the Research Electronic Data Capture (REDCap) platform [13,14], and weblinks were emailed to eligible participants. At T2, we sent a follow-up email to the entire T1 sample inviting them to complete the second assessment. Undelivered email invitations indicated that a participant no longer worked at MSH. Anonymity was preserved using approximate deterministic linkage methods applied to participants’ self-generated research codes. Linked responses were those with exact code matches and those within one generalized Levenshtein edit [15] and 4 out of 5 five matching demographic variables [2,16]. Participants were eligible to receive prizes via raffle by filling out a separate unlinked form.

Individuals eligible to participate in both surveys included MSH FHCWs who directly cared for COVID-19 patients during T1, either as part of their standard scope of practice or due to T1 redeployment assignment. This sample consisted of attending-level physicians and house staff from various departments, patient-facing nurses and nurse practitioners, physician assistants, chaplains, clinical psychologists, social workers, and dietitians. The Institutional Review Board (IRB) at the Icahn School of Medicine at Mount Sinai approved this study.

### 2.2. Mental Health Treatment Utilization

Mental health treatment status during the COVID-19 pandemic was assessed at T1 and T2 using the following question: “Are you currently receiving treatment for a mental health condition?” FHCWs were classified into four groups based on responses to these questions: (No Treatment: Endorsement of “No” at T1 and T2; Stopped Treatment: Endorsement of “Yes” at T1 and “No” at T2; New-onset Treatment: Endorsement of “No” at T1 and “Yes” at T2; and Continued Treatment: Endorsement of “Yes” at T1 and T2).

### 2.3. Study Instruments

Appendix A describes all the variables assessed at T1 and T2 that were examined as potential predictors/correlates of mental health treatment utilization. These variables included demographic and occupational characteristics, COVID-19-related personal/occupational stressors, coping strategies/restorative behaviors, and psychosocial characteristics [1,17,18,19,20,21,22,23,24,25,26,27,28,29,30,31].

### 2.4. Data Analysis

Data analyses proceeded in two steps. First, chi-square and analyses of variance were conducted to compare sociodemographic, occupational, COVID-19, and psychosocial variables by mental health treatment status (no treatment, stopped treatment, new-onset treatment, continued treatment). Bonferroni-corrected pairwise contrasts were computed to identify significant between-group differences. Second, three multivariable binary logistic regression models with forward likelihood ratio estimation were conducted to identify variables associated with new-onset vs. no treatment; continued vs. no treatment; and continued vs. stopped treatment; variables associated with mental health treatment status at the *p* < 0.05 level in bivariate analyses were entered into these analyses. To test for possible interactions of variables that predicted mental health treatment, all 2-way variable combinations were entered into a second step of the regression models.

## 3. Results

Of the 6026 presumed FHCWs who were sent the survey at T1, 3360 completed the survey. Of the T1 survey respondents, 781 respondents were excluded because they did not endorse frontline clinical responsibilities. 787 participants completed the survey at T2. In this sample, 780 responded to the mental health service utilization questions.

In the sample, 19.1% (*n =* 149) of FHCWs reported utilizing mental health services at T1; within this group, 11.4% (*n* = 89) reported continuing care at T2, 4.2% (*n* = 33) initiating care at T2, and 3.5% (*n* = 27) reported being in care at T1 but not at T2.

As shown in Appendix A, at T1, mental health treatment groups differed with respect to race/ethnicity, presence of children in the household, profession, history of a mental health diagnosis, perceived preparedness, positive screen for MDD, GAD, and burnout, relationship and work difficulties, and distress related to systemic racism and racial disparities in COVID-19 outcomes. Group differences were also observed for positive emotions, perceived resilience, and protective psychosocial characteristics at T1, as well as sleep hours and the number of self-sufficient and avoidant coping strategies used at T1. Table 1 shows results of multivariable logistic regression models predicting new-onset and continued mental health treatment utilization.

Relative to FHCWs who did not utilize mental health treatment, those who initiated treatment during the pandemic were more likely to have a history of a mental health diagnosis, report greater distress related to systemic racism during the pandemic, and score lower on measures of feeling interested, perceived resilience, and optimism at T1. A significant interaction between mental health diagnosis history and perceived resilience at T1 was also observed (OR = 0.47, 0.23–0.94, *p* = 0.033; Figure 1); among FHCWs with a history of a mental health diagnosis, those with greater perceived resilience at T1 were less likely than those with lower perceived resilience to report engagement in mental health treatment.

Relative to FHCWs who did not utilize mental health treatment, those who continued treatment were more likely to be White, non-Hispanic, have a history of a mental health diagnosis, report greater distress related to systemic racism during the pandemic, and score lower on a measure of perceived resilience at T1. No interactions were significant, all *p*’s > 0.5.

Relative to FHCWs who ceased mental health treatment between T1 and T2, those who continued mental health treatment were more likely to have a history of a mental health diagnosis, report greater relationship difficulties at T1, and were less likely to use positive reframing and planning coping strategies at T1. None of the interactions were significant, all *p*’s > 0.15.

Figure 2 shows the supportive resources that FHCWs identified as being helpful in reducing pandemic-related stress. The most endorsed resources were reliable access to personal protective equipment (PPE), access to scrubs, food provisions, clear system-wide communications, and access to healthy snacks at work.

## 4. Discussion

This study examined the prevalence and correlates of different courses of mental health service utilization across two time points during the first year of the pandemic in COVID-19 FHCWs in New York City. Overall, we found that 19.1% of FHCWs self-reported utilization of mental health services during the study period. This result is somewhat higher than the 13.1% utilization found in a prior study of physicians [4], and may be attributable in part to our sample comprising a broader range of FHCW employment roles, some of which have higher rates of adverse mental health outcomes (e.g., nurses [1]).

Multiple T1 variables predicted the initiation and continuation of mental health services. Having a pre-pandemic history of a mental health diagnosis (a need-based factor in the AMBHSU) was associated with a greater likelihood that FHCWs would initiate or continue care; and greater positive emotions and perceived resilience were associated with a lower likelihood of service use. FHCWs identifying as White and non-Hispanic (predisposing factor) were more likely to continue care. This finding could be attributed in part to differences in access to resources, lack of time to find or engage in treatment, or mental health stigma between racial/ethnic groups in the sample [32]. Importantly, the barriers to mental health care that FHCWs may experience in relation to their marginalized status(es) requires further study.

Of note, relative to those who continued care, those who ceased care by T2 were more likely to endorse using positive reframing and planning as coping strategies at T1. One possible interpretation of this finding is that FHCWs who ceased care may have applied coping skills learned in a successful time-limited treatment (e.g., cognitive therapy) in managing pandemic-related stressors.

We also observed an interaction between prior MH history and perceived resilience, such that FHCWs with a mental health history and with greater resilience were less likely to newly engage in care relative to those with lower resilience. A possible interpretation of this finding is that those with a greater perceived resilience (a need-based factor in the AMBHSU) may feel that they have skills to cope with their prior mental health symptoms, and therefore didn’t feel the need to seek treatment during the study period. Therefore, this finding underscores the importance of interventions to bolster resilience in FHCWs with prior mental health diagnoses [33]. Prior work with veterans has also shown resilience and optimism to be associated with lower likelihood of mental health service use [34]. To our knowledge, this study is the first to demonstrate a link between distress related to systemic racism (a need factor in AMBHSU) and mental health service use in FHCWs. Further research is needed to examine the adverse impact of contextual factors in FHCWs, and our work brings attention to the need for tailored interventions for FHCWs feeling the impact of racial injustice.

Placing service use in further context, descriptive data revealed that a small subset of respondents (<10%) reported that mental health services were helpful in managing pandemic-related stress, with the larger majority reporting that basic needs and safety-related supplies were helpful. An implication of this finding is that in addition to increasing access to formal mental health services, health care systems preparing for future disasters should make ample investment in basic needs and safety-related supplies and system-wide policies to distribute basic-need and safety supplies in the event of an emergency.

### Limitations

The limitations of this study are notable. First, our self-reported measure of mental health service use does not permit distinctions among types of services (e.g., psychotherapy and/or medication management; in-person vs. virtual care), frequency, duration, or measurable impact of service engagement. As has been done in studies of veterans [34,35] and physicians working during the first year of the pandemic [4], further studies of mental health treatment services in FHCWs should consider administrative/claims data to assess a broader range of information regarding mental health service utilization. Second, we studied only FHCWs, and do not report on data concerning those individuals in research, support, or administrative roles within the healthcare system, many of whom also have heightened levels of distress [36]. Additionally, relatively few participants started or discontinued care at T2 (*n*s = 33 and 27, respectively), which led to small cell counts within predictors for those groups and potentially underpowered analyses. Finally, since we only used one question to measure distress related to systemic racism (”To what extent have you felt emotionally affected or distressed by systemic racism highlighted by recent events across the country?”), it is unclear whether respondents are specifically referring to systemic factors (e.g., political, economic, institutional policies, etc.) or more visible examples of interpersonal conflict (e.g., hate crimes) in their responses. Also, considering the small cell size of those who endorsed racism-related distress in this sample, the conclusions that can be drawn from this question are limited.

## 5. Conclusions

Notwithstanding the limitations noted above, results of the current study highlight several areas of potential clinical intervention and provide ancillary support for areas of future study. First, specific efforts should be made to engage FHCWs who identify as non-White and/or non-Hispanic, because they may be less likely to engage in care even in the presence of substantial need [32]. Future research should further assess how that predisposing and need-based factors, such as stigma and distress related to systemic racism, respectively, contribute to or inhibit marginalized FHCWs seeking mental health services. Psychoeducational videos that have included personal testimonials have been shown to be a successful avenue for increasing treatment seeking intentions in FHCWs [37], and may help reduce stigma. Second, several potentially modifiable psychological factors, spanning positive emotions, perceived resilience, and coping skills, emerged as significant predictors of service use. Further research is needed to more carefully examine how these factors may influence help-seeking over time, and whether modifying them through prevention and treatment efforts may help mitigate distress.

## Figures and Tables

**Figure 1 ijerph-20-05326-f001:**
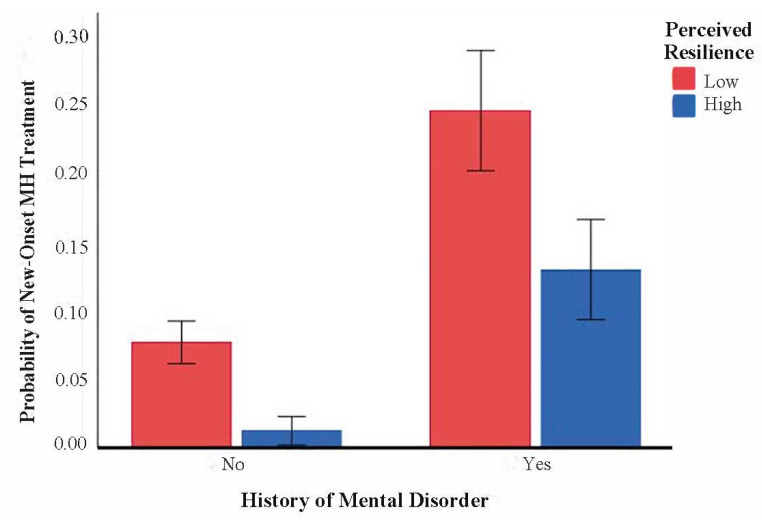
Interaction between perceived psychological resilience (as measured by the CD-RISC 2) and mental health history in the prediction of new-onset mental health (MH) treatment in HCWs. Error bars represent 95% confidence intervals.

**Figure 2 ijerph-20-05326-f002:**
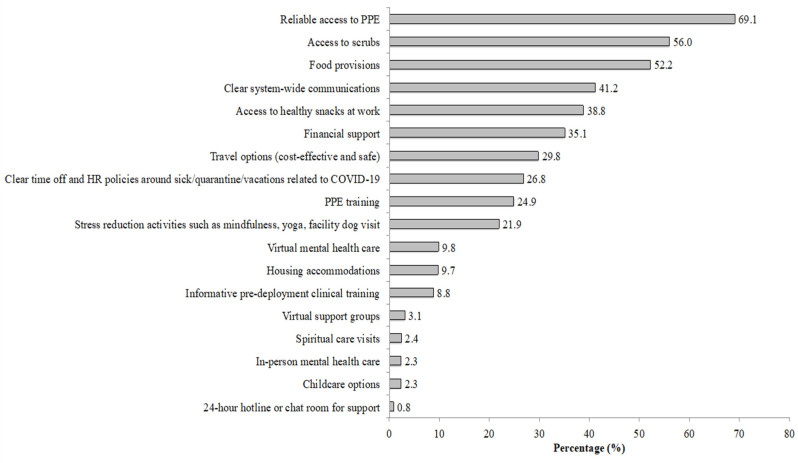
Support resources that COVID-19 frontline health care workers found helpful in reducing pandemic-related stress Note: participants were asked, “Which of the following support resources (if any) did you use to reduce some of the stress around COVID-19?” HR = human resources; PPE = personal protective equipment.

**Table 1 ijerph-20-05326-t001:** Predictors of mental health treatment among COVID-19 frontline health care workers.

**New-Onset vs. No Mental Health Treatment**		
Nagelkerke R^2^ = 0.28		Odds Ratio (95% CI)
	History of mental health diagnosis (need)	5.29 (2.24–12.48) ***
	Distress related to systemic racism (need)	2.17 (1.47–3.22) ***
	Positive affect: Interested (need)	0.57 (0.38–0.87) **
	Perceived resilience (need)	0.68 (0.48–0.97) *
	Optimism (need)	0.76 (0.57–0.99) *
**Continued vs. No Mental Health Treatment**		
Nagelkerke R^2^ = 0.51		
	White, non-Hispanic race (predisposing)	2.99 (1.53–5.83) **
	History of mental health diagnosis (need)	32.29 (17.26–60.38) ***
	Distress related to systemic racism (need)	1.35 (1.06–1.74) *
	Perceived resilience (need)	0.73 (0.58–0.93) **
**Continued vs. Ceased Mental Health Treatment**		
Nagelkerke R^2^ = 0.33		
	History of mental diagnosis(need)	4.49 (1.62–12.39) **
	Relationship difficulties (need)	1.03 (1.01–1.06) *
	Positive reframing (need)	0.26 (0.07–0.90) *
	Planning (need)	0.19 (0.04–0.86) *

Note: The scales used to measure each predictor variable are listed in Appendix A. For each variable, the best-fit domain of the ABMHSU framework (“predisposing”, “enabling”, or “need”) is in parentheses. * *p* < 0.05, ** *p* < 0.01, *** *p* < 0.001.

## Data Availability

Due to their sensitive nature, the data presented in this study are restricted by the corresponding author and are not publicly available.

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
