# Peer review of "Predictors of Mental Health Service Utilization among Frontline Healthcare Workers during the COVID-19 Pandemic"

_ijerph, 2023, doi:10.3390/ijerph20075326_

Round 1

Reviewer 1 Report

General concept comments

Authors aimed to assess factors associated with initiation, continuation, or cessation of mental health services among FHCWs. Specific hypotheses included testing whether: 1) demographic factors (female gender, prior mental health history, and current psychiatric symptom burden) would emerge as the strongest predictors of service use across study period, 2) distress related to systemic racism would be associated with an increased likelihood of service use, and to 3) describe services that FHCWs perceived as most helpful in reducing stress during the pandemic, spanning from basic and safety needs to mental health services. Hypotheses are clear and relatively straightforward based on data collected. 

One concern, partially referenced in the limitations section as well, is sample size for T2 participants continuing care (n=33) and discontinuing care (n=27) and the ability to draw meaningful conclusions from sample size. Similarly, per supplemental table 2, across service utilization groups, only up to three individuals endorsed distress related to systemic racism or treatment disparities. It is not feasible to draw even preliminary quantitative data from this sample size. 

Please see the following for additional concerns related to the measure of systemic racism. Authors ask two questions to measure distress related to systemic racism and race-related treatment disparities during the COVID-19 pandemic:

1. Distress related to systemic racism: To what extent have you felt emotionally affected or distressed by systemic racism highlighted by recent events across the country? 

2. Distress related to racial disparities in COVID-19 health outcomes: To what extent have you felt emotionally affected or distressed by the racial disparities in COVID-19-related health outcomes?

Systemic racism, as it is so embedded into the structures of society, is often outside of individuals conscious awareness and is therefore more difficult for individuals to recognize and measure. Without providing more context for participants to answer items inquiring about racism-related stress, it is unclear whether responses are referring to systemic racism (e.g., political, legal, economic, health care, school, criminal justice systems, policies, entrenched beliefs, institutional practices), or more visible forms of interpersonal (e.g., hate crimes) and internalized racism. Consider shifting your language to reflect the measurement of racism at the interpersonal/ individual levels or clarify how systemic racism is measured. 

The authors mention the murder of George Floyd in the article as a driving factor for discussion of systemic racism in the US during the COVID-19 pandemic. While there is an established relationship between police violence against Black men and the association between racially motivated police killings and health outcomes among Black communities, the authors of this study (at least based on supplemental materials provided) do not assess this relationship in the current article. It is recommended, therefore, that authors 1) provide additional context and information for the importance of studying systemic racism and how this applies to FHCWs and 2) remove the explicit reference to George Floyd and speak more generally about increased rates of racial health inequities and its impact on FHCW, etc. 

Significant contributions and strengths of this study are also the areas that would benefit from improvement. The use of the ABMHSU provides a solid framework for understanding potential risk factors. That said, it may benefit readers for additional information about the model, specifically differentiation of factor categorization (and the significance of categorization). Additional information may also be useful regarding the utility of this framework for understanding service utilization and, more broadly, HCW wellness.

A strength of this article is the inclusion of reference to the importance of understanding FHCWs experience with multiple marginalized statuses (i.e., female gender identity, POC, and individuals with mental health concerns). The article would benefit from additional description of the importance of studying multiple marginalized groups and the impact of stigma and discrimination (whether at the systemic level or individual level) on health outcomes. At present, the inclusion of aspects of diversity appears as an afterthought, rather than an intentional decision to study this aspect of intersectional experience. Additional suggestions include adjusting language around mental health to state mental health diagnosis rather than mental disorder and implementing person-first language throughout the text. Further, when referring to racial-minority individuals, ensuring that discussion includes an understanding that the experiences of individuals drastically differed during the pandemic, whether referring to police violence against Black men or the increase in violence toward Asian and Asian American individuals.

Results: As stated earlier, the main concern is the small sample sizes in most of the treatment categories (and much smaller cell sizes when examining important factors such as distress related to racism) resulting in low power to detect real differences and having unstable effect sizes. It is not clear that using multivariable modeling where additional variables were adjusted for would even be appropriate with such small sample sizes. The small sample sizes also resulted in very uneven group comparisons when the No Tx group was involved. In addition, there are some group comparisons that were not conducted (or the data was not reported). It would be interesting to further examine the “No MH Tx” at baseline as compared to Any Treatment at baseline to determine what factors were associated with any treatment at the same point in time. This would also contribute to the discussion of barriers to care among those who were elevated on MH symptoms but were not engaged in any treatment at the time. This could also be conducted at T2 (any tx vs no tx) so that there is temporal consistency (and potentially higher sample sizes). The overall sample size is never mentioned and there is no discussion of what the sample size at T1 was before it was reduced to include only those who also completed the T2 measures. Again, perhaps using all respondents at T1 would have increased power for cross sectional analysis at T1. A discussion of attrition, however, would need to follow. Finally, interaction effects must be interpreted with caution with these very small sample sizes. This needs to be reiterated in the limitations. Additional limitations are the use of non-validated scales for measurement of key factors such as those involving racism-related distress.

Please consider adjusting for deployed vs not deployed status of FHCWs as we know that literature points to deployment as a risk factor for mental health difficulties. 

Discussion: It would be helpful to provide a stronger explanation of why those with a history of mental health difficulties and low resilience were more likely to seek treatment- the argument could easily be made that treatment would be a resource used by those with higher resilience. It would also be beneficial to include information on future directions. 

Specific comments  

1. Lines 47 and 57-58. Line 47 lists employment as an example of an ‘enabling’ factor, however lines 57-58 classify “extreme occupational stressors” as predisposing factors. It appears the AMBHSU categorizes employment as a social structure within predisposing factors; please clarify.

2. Line 59. Please add a citation for the reported increase in racially motivated violence during the pandemic specifically (in addition to the Qureshi et al., 2021 article).

3. Line 60-61. In the context of the current article, referencing the murder of George Floyd, while incredibly significant, is a bit out of place. It may be more relevant to include a reference to the increase in racially motivated violence specifically targeting Asian American individuals during the COVID-19 pandemic, in addition to the citation noting the disproportionate mortality burden among POC. 

4. Line 69. “Female gender” is listed as a potential predictor of service use. Please clarify whether participants were asked about gender identity or sex assigned at birth (i.e., male, female, intersex) and adjust accordingly.  

5. Lines 72-73. Please clarify classification of “distress related to systemic racism” as a predisposing factor. Per the AMBHSU, while race, a demographic predisposing factor, and racism, a socio-cultural factor predating health outcomes would be considered predisposing factors, stress/distress related to racism appear to fall within the perceived/evaluated needs factors for participants. Please clarify the inclusion of distress as a predisposing factor rather than a needs factor.

6. Lines 102-104. Please clarify whether participants were also asked at T1 how long they have been in treatment, as service utilization may not be related to the pandemic.

7. Table 1. Like previous comments related to factor categorization, please clarify decision to categorize positive affect, perceived resilience, optimism, planning, and positive reframing as need factors rather than predisposing (personality) factors.

8. Table 1. Provide additional variable labels to aid understanding of the table.

9. Lines 186-188. Please add citations to hypothesized explanations for racial differences in continuing treatment, given that “access to resources, lack of time to find or engage in treatment, or mental health stigma between racial/ethnic groups” were not directly assessed in current study.

10. Lines 226-228. A gap remains between what was studied in this article and the conclusions drawn from the data, specifically that racial differences in service utilization stem from racial minority individuals holding “greater mental health care stigma.” Agreed, however, stigma was not explicitly measured in this article, therefore inclusion of this conclusion would require a citation and prefacing results of the current study provide ancillary support for future studies to explore the association between stigma and mental health utilization among racial minority individuals.

11. Supplemental Document: Track change comments are still visible.  

Reviewer 2 Report

Manuscript ID: 2208874 is Predictors of Mental Health Service Utilization Among Front-line Healthcare Workers During the COVID-19 Pandemic. The authors have taken the prevalence and correlate of factors associated with self-reported mental health service use in a longitudinal cohort of frontline health care workers  (FHCWs) providing care to patients with COVID-19 throughout. Overall, the article focuses on the burning issue, and the quality is good; however, there is always room for improvement. I have highlighted some critical points the authors need to address to enhance the academic quality of the manuscript.

1.     The authors have calculated short, and long-run relationships between different sets of a prior mental health history, those with greater perceived resilience were less likely to initiate treatment. However, the abstract is silent about these relationships. The authors must write about the long-run relationship because the policy always depends on long-run dynamics.

2.       The introduction section tells us about how The COVID-19 pandemic brought unprecedented amounts of stress to frontline healthcare workers. However, it is silent about the main contribution of this research to the current stream of the literature. Besides, the authors also need to highlight some of the major findings along with the contribution at the end of the introduction section.

3.       I would like the authors to comment on the Data collected via two anonymous surveys of FHCWs working at Mount Sinai Hospital (MSH), an urban tertiary care hospital in NYC. The first survey was administered during the middle, and the downward slope of the initial pandemic peak in NYC in April-May 2020 is the author's results or international Data to support their arguments

4.       Authors need to pay serious attention to many grammatical and sentence errors because sometimes it becomes hard to follow the manuscript.

5.       The authors have mentioned about MDD and GAD, and just provided a few references; they need to provide more information about these concepts and let us know what kind of situations and logistic are suitable for them.

6.       The estimation strategy is fine; however, the authors need to complete atleast one set of the predictor variable; for instance, the variable for frontline healthcare workers in terms of the travel of mental health workers needs to be a complete set that the authors have identified in table 1.

7.       The authors need to explain the information in figure 1 and figure 2, as given in the estimation section; it is needed to explain what is the threshold value and whether the author's estimation of the bounds test needs the required criteria.

8.       The conclusion and implication section is well-written. However, the author is required to provide more policy implications. Besides, there is always some research limitation; the authors must add the limitations. Further,  what are the future perspectives and directions to extend this research? It would be great if the authors could provide directions to the new researchers.

Round 2

Reviewer 1 Report

I recommend accepting the revised manuscript, however, the fact that the results should be interpreted with caution based on the small sample sizes should be restated in the conclusion. 

Reviewer 2 Report

Thank you for giving a detailed response to the reviewer and for correcting previously reported concerns.